# Nested Text Labelling Structures to Organize Knowledge in AI Applications

## Abstract

Scientific literature has emerged to advance annotation frameworks incorporating multi-fragment and multi-assessor labelling protocols alongside contextual data. The application of such rigorously defined, expert-driven text annotations provides a foundation for developing machine learning models capable of performing automatic text markup. The paper aims to identify the knowledge representations suitable for both human annotators and machine learning processes, as well as various task types. Experience gained through a number of applied projects and research studies has shown that the answer is not that simple. We propose a multi-level approach to the data models used for text annotation. Given its applicability for tasks involving context, multi-assessor labelling, and the extraction of subjective textual categories, this paper delineates its conceptual and logical foundations, alongside the associated cases. The proposed framework comprises three nested data models, each distinguished by its level of complexity. The relational representation of textual annotations is flexible enough for a variety of annotation scenarios. It supports named entity recognition, relation extraction, semantic analysis, co-reference resolution, frame semantics, multi-span matching, etc.— at least 17 types of tasks whose inputs and outputs have fundamentally different structural complexities. The framework includes a core model, an extended set of entities, and their relations. The same dataset can be related to various tasks of significantly different types. The broad applicability of our framework is supported by the survey of 21 datasets and related tasks found in more than a thousand publications. The proposed methodology extends the scope of structured text annotation, advances the standardisation of content analysis procedures, and facilitates solutions for a broader spectrum of natural language processing tasks.

## 1 Introduction

Knowledge can manifest in various forms, including ideas, methodologies, and raw data. Yet knowledge analysis automation necessitates formal representations. In turn, automation allows us to organize research more efficiently and answer more complex questions. Scientific literature for Digital Humanities demonstrates artificial intelligence applications underpinned by sophisticated annotation frameworks, which integrate multi-fragment and multi-assessor labelling protocols enriched with contextual metadata. Illustrative applications include the tagging of nuanced semantic errors — such as identifying flawed causal reasoning — in secondary-level student essays, alongside the annotation of inherently subjective constructs, including universal human values.

This research seeks to identify optimal data models for various content analysis scenarios that serve both human annotators and machine learning processes, addressing the fundamental trade-off between data model convenience and expressiveness.

The proposed text markup framework comprises three nested data models, each exhibiting distinct levels of intricacy. In this work, nested data models can also be referred to as hierarchical or multi-level. The framework emphasizes hierarchical and relational representations of text annotations.

The development of our multilevel system of data models was informed by two principal requirements. Firstly, the system needed to be adaptable to a diverse spectrum of text annotation tasks, such as named entity recognition, relation extraction, semantic analysis, co-reference resolution, frame semantics, and multi-fragment search. To demonstrate the broad applicability of the proposed data

models, we developed and applied a pipeline for identifying recurring problem statements within the field of text annotation. An extensive semi-automated literature review, conducted using this pipeline, has identified seventeen distinct task types characterised by fundamentally different levels of structural complexity in their inputs and outputs. The empirical validation of the model hierarchy involved mapping these tasks from twenty-one datasets across its different levels.

Second, the annotation process itself must be intuitive for a domain expert. The starting point of engineering the nested models is that semantic annotation of a text can be achieved through four elementary operations: highlighting a section of text, categorizing it, linking spans if necessary, and providing commentary on the section or its relationship if necessary. These four operations appear to be sufficient to analyse media narratives or formalize various issues related to understanding the semantic nature of a text (Rink et al., 2024).

The study allowed us to identify 17 types of tasks related to labelling texts. The employment of the proposed model hierarchy during development, coupled with training on diverse task types, is posited to yield a new generation of machine learning models. These models would be characterized by greater usability and a broader range of applications.

## 2 RELATED WORK

During recent decades, scholars engaged in textual analysis — particularly within the social sciences and humanities — have devised ever more sophisticated annotation schemes and data models. An established academic consensus, as evidenced by comprehensive literature surveys, acknowledges the proliferation of novel representational frameworks. This development constitutes a paradigm shift from simple markup conventions towards increasingly granular, structured, and semantically expressive formats. A partial trajectory of this evolution is chronicled in Portier et al. (2012), who traces its origins to the Standard Generalized Markup Language Fraser et al. (1986).

A parallel transformation unfolded within natural language processing, catalysed by the rise of multi-task learning and models trained on extensively diverse datasets. Foundational research on architectures such as BERT established that performance improvements were increasingly attributable not to novel structural components, but to strategic alterations in the composition and interplay of pre-training tasks. This principle is further substantiated by subsequent iterations of BERT and related transformer-based systems, which underscore that the selection and integration of training tasks have emerged as primary determinants of model capability and generalisation (e.g., Devlin et al. (2019); Raffel et al. (2020); Wang et al. (2018)).

A further cardinal driver of recent progress in artificial intelligence has been the deployment of large-scale, heterogeneous corpora. An established empirical consensus holds that model robustness and cross-domain adaptability are significantly enhanced when training data encompass a broad spectrum of linguistic registers, genres, and real-world application scenarios. Consequently, data diversity has emerged as a factor of importance equal to architectural innovation in the development of state-of-the-art natural language processing systems.

Collectively, these developments indicate that advancing artificial intelligence methodologies for text processing — particularly within domains such as the social sciences and humanities — necessitates addressing a twofold challenge. Scholars must devise models capable of accommodating multiple task types whilst simultaneously curating comprehensive datasets that encompass varied subject domains and divergent annotation practices. The integration of multi-task modelling with large-scale, heterogeneous data thus represents a critical frontier for constructing the next generation of AI tools dedicated to text-centred research.

In contemporary research, new datasets and benchmarks are released at an accelerating pace. For this study, we have assembled a collection of datasets spanning a broad spectrum of domains such as the humanities Rink et al. (2024), medicine Gurulingappa et al. (2012), edtech Levikin et al. (2025), chemistry Krallinger M. (2017), and linguistics ( Gordeev et al. (2020), Luan et al. (2018), Bos et al. (2017), Loukachevitch et al. (2023)). This methodological approach to dataset curation and examination aligns directly with the theoretical and practical evolution of the natural language processing field, as previously outlined. In accordance with this paradigm, we identified a corpus of 21 datasets listed in Tab. 1. Subsequent analysis reveals that a significant proportion of the datasets

within the assembled corpus demonstrate multifunctional utility across several tasks and exhibit considerable cross-disciplinary breadth.

The considerations outlined above for developing next-generation AI models necessitate the consolidation of diverse existing datasets. This integration, in turn, requires the formulation of a universal data model capable of consistently accommodating all task types associated with each constituent dataset.

A prominent contemporary model of markup representation, which continues to evolve and finds extensive application within social sciences and humanities scholarship, is Text Encoding and Interchange Burnard et al. (1994). A principal limitation of the Text Encoding and Interchange (TEI) framework is the inherent tension between semantic expressivity and the challenge of disambiguating text spans that can be associated with multiple, often conflicting, semantic interpretations. This fundamental constraint necessitates the development of new, universal data models capable of fulfilling the multifunctional requirements previously outlined.

## 3 Methodology: Text Annotation Scenarios

### 3.1 Extensive Review Pipeline

A systematic methodology is developed to catalogue common text annotation scenarios, with the objective of creating a system of data models possessing wide-ranging utility. An important consideration is that the same dataset can be employed to study multiple tasks, which can differ in terms of both their semantic content and formal signatures.

During the initial phase of the research, it became apparent that a manually conducted literature review would be incapable of achieving the necessary level of completeness. Existing methods proved inadequate, primarily due to limitations in scalability and coverage. To address this, a novel semi-automated literature review pipeline was developed.

The adopted methodology comprises the analysis of expert-curated datasets via the Semantic Scholar API to identify relevant citing publications. A subsequent workflow, employing a Large Language Model (LLM), is designed to systematically extract and categorise annotation tasks from this corpus. The resultant taxonomy of common text annotation tasks provided a basis for formulating generalised annotation models. This framework was further refined by establishing a comprehensive mapping between a wide spectrum of tasks and the model's hierarchical levels (Table 1). Further details of the pipeline are provided in the Appendix F.

### 3.2 Task Catalogue

We have analysed 21 datasets through manual and semi-automated processing of over a thousand associated publications and identified 17 types of problem statements. They are characterised by fundamental differences in the structural complexity and semantics of their inputs and outputs. The relationship between datasets and these tasks is mapped in Table 1.

Given the considerable variation in problem statements and data structures across these tasks, we propose a classification system based on three levels of dataset complexity, which are described as follows.

**I. Solid spans**    The tasks, where we identify the spans in a given text and label the spans (or the text wholly) with a single tag, are well-researched. Named entity recognition with multi-class classification is an illustration. Here, spans are entities and are of small or medium size. An example is the Kaggle NER corpus dataset (Bos et al., 2017).

**II. Elements / Combinations of spans**    Sometimes, a problem statement involves finding so-called *elements* containing a number of spans. In content analysis, the annotation of multiple spans has been shown to enhance the identification of human values in textual data (Maysuradze et al., 2024). In addition, we can label spans and elements with multiple tags, with Relationship extraction (RE) and co-reference analysis being the main examples.

Relation extraction can be used in a variety of applications, including information extraction, question answering, and natural language processing. MERA (Fenogenova et al., 2024), ADE (Gurulingappa et al., 2012), NEREL (Loukachevitch et al., 2023), RURED (Gordeev et al., 2020), RuSuperGlue (Shavrina et al., 2020), SCIERC (Luan et al., 2018), SemEval 2010 task 8 (Hendrickx et al., 2019) are some examples.

Co-reference resolution is the process of finding and linking words or phrases in a text that all refer to the same entity, such as a person, place, or thing. The examples may be CONLL 2012 Ontonotes (Pradhan et al., 2013), SCIERC (Luan et al., 2018). RuSentNE (Golubev et al., 2023) is another example of a model that uses not only entity tags but also sentiment tags to label its output.

**III. Extensions**   Annotations of documents can be created by multiple assessors and may contain comments and references to a context. In addition, spans may have an unlimited size and contain a wide range of meanings. Elements may have complex structures, such as frames and multi-fragments. Frame structures consist of spans that play fixed roles. Multi-fragments (Maysuradze et al., 2024) are composed of connected phrases that are scattered throughout the text and have a specific meaning. They can also be used as an alternative to a single, continuous phrase when the frame structure is insufficient. This is the case when multi-span phrases do not have fixed positions or when those positions may be repeated an unlimited number of times. Datasets, such as UpGreat READ//ABLE (identifying semantic errors in school essays) (Levikin et al., 2025) and Human Values (detecting human values and their sentiments in social media texts) (Rink et al., 2025), demonstrate the level of complexity.

ADE (Gurulingappa et al., 2012), CONLL 2012 OntoNotes (Pradhan et al., 2013), and Kaggle NER (Bos et al., 2017) corpora also maintain several annotations per document. However, we do not have information on the annotators in these cases. These instances are relatively rare compared to the two datasets we indicated. Additionally, multi-spans are present in NEREL (Loukachevitch et al., 2023) and RURED (Gordeev et al., 2020), also very rarely.

# 4  NESTED DATA MODELS

Tasks, topics, and the inherent complexity of annotation exhibit significant diversity, rendering any single conventional data model insufficient for successful knowledge formalisation. To address this shortcoming, a system of nested data models has been developed. In constructing this framework, we have been leveraging insights from existing datasets. Our approach involved the unification of datasets and the development of nested data models that align with distinct tiers of task complexity. Consequently, this structure facilitates the selection of an appropriate AI model for any specific dataset, ensuring a more efficient analytical approach. It is appropriate to repeat here that the same dataset can be associated with tasks of different levels of data model complexity.

To visualize the annotation structures associated with various problem statements, Fig. 1 provides an Entity-Relationship diagram in Crow's Foot notation (Everest, 1976). The following subsections provide a brief description of the different levels of our system of data models; more detailed descriptions may be found in the appendix. Additionally, we have proposed standard transitions between hierarchy levels, which are described in detail in Appendix D.

## 4.1  LEVEL OF SPANS

At this tier, annotations are comprised solely of text spans (continuous fragments). Each span is assigned a single tag (first sub-level) or a set of tags (second sub-level). This structure pertains to multi-class classification and named entity recognition tasks.

Furthermore, a document is associated with a single annotation (first sub-level) or may possess zero to multiple markups (second sub-level). The level of span structure encompasses four distinct entities:

- *Document* is any object (file) that supports conversion to a text string

- *Markup* is a markup of document with spans

- *ESpan* corresponds to a substring of document (Elementary span)

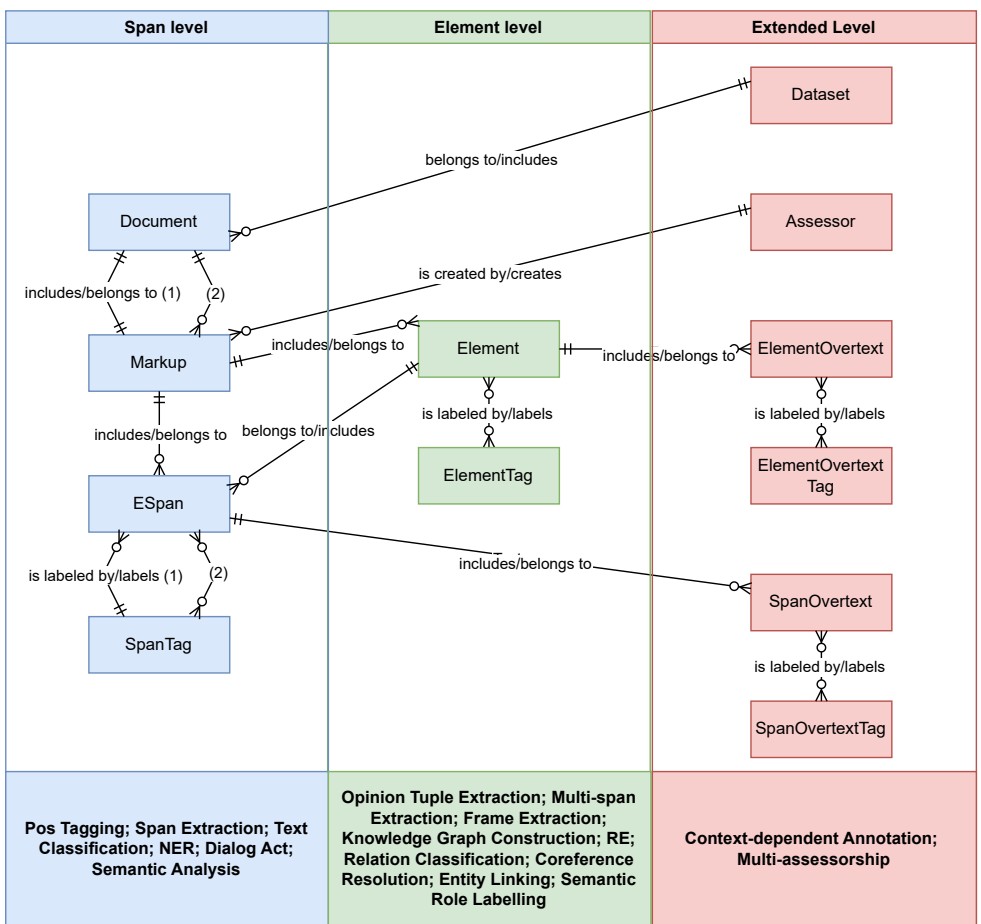

Figure 1: ER-diagram for the proposed system of nested data models.

- *SpanTag* is used to tag spans

Level of spans also includes 3 relationships between its entities. As we mentioned, there are two different versions of the relation between ESpan and SpanTag and the relation between Document and Markup; they are marked by (1) and (2) in Fig. 1, respectively, to the first and second sub-levels.

- *SpanTag — ESpan* The relation has the following properties: each span is labelled with a single span tag (0 or more span tags for second sub-level). Span tag may label 0 or more spans.

- *ESpan — Markup* The relation has the following names: "Span belongs to Markup" and "Markup includes Span". The relation has these features: A Span can belong to only one Markup. A Markup may contain zero or more Spans. If a Markup contains zero Spans, it means that the assessor did not find any Spans in the document.

- *Markup — Document* The relation has the following names: "Markup belongs to Document" and "Document includes Markup". The relation has the following features: Markup corresponds to only one specific Document, and a Document have only one Markup (a Document may have zero or more Markups for second sub-level).

Two different spans may correspond to the same substring of the document's text. So labelling of a span by several span tags on the second sub-level may also be realized on the first sub-level: several spans with the same substring are labelled by these tags one by one. But we should remember that

the correspondence between a single span and two different tags, and the correspondence between two spans with the same substring and these two tags, may have different meanings in the general case.

In addition, the presence of 0 tags for a fragment (empty set of tags on the second sub-level) in most datasets could have some specific meaning: all the extracted fragments could be marked with the same tag, so the tag was omitted.

The data model described in this subsection is a traditional one and can be found in many different datasets. As described earlier, in this case, each fragment is associated with exactly one tag and each document has a unique markup. We intentionally added multiple tags to the fragments and multiple markups to the documents on the second sub-level. This allows us to work with multiple assessments because documents from different datasets usually correspond to a single markup (as in the first sub-level), but several of them may have different markups that correspond to the same text. The second sub-level helps us solve this issue.

Examples are given in Appendix A.

## 4.2 LEVEL OF ELEMENTS

At this level, we have two additional entities: Element and ElementTag. In addition, spans now belong to elements, and the name ESpan can be interpreted as the "element's span". As a result, definitions of these concepts change as well.

Therefore, at this level, annotations contain elements, and each element consists of spans. In addition, a single span can be labelled with an arbitrary number of tags (like in the second sub-level of the level of spans). A typical example of an element at this level is the frame structure described above, where each span is labelled with a unique tag. In the RE task, an element with this feature can be interpreted as a relation.

- *Markup* is a markup of document with elements (instead of spans)
- *Element* is a set of related spans
- *ElementTag* is a label used to tag elements
- *ESpan* corresponds to a substring of document in specific element (Element span)

There are 3 new relations (*Element — Markup*, *ESpan — Element*, *ElementTag — Element*):

- *ElementTag — Element* Relation has the names: ElementTag labels Element, Element is labelled by ElementTag. Relation has the features: the same ElementTag can label 0 and more Elements; Element can be labelled by 0 and more ElementTags (It is allowed to have Element without ElementTag)
- *ESpan — Element* Relation has the names: ESpan belongs to Element, Element includes ESpan. Relation has the features: ESpan can belong only to one Element, so it can be considered as *Element item*; Element may include 0 and more ESpans
- *Element — Markup* Relation has the names: Element belongs to Markup, Markup includes Element. Relation has the features: Element can belong only to one Markup; Markup may include 0 and more Elements (having 0 Elements means that assessor hasn't found in Document any Elements and thus hasn't found any ESpans)

The 1-to-many relationship *ESpan — Markup* is also presented in this data model, but as a composition of relations *ESpan — Element* and *Element — Markup*. Besides that, ESpan can still be labelled by 0 or more SpanTags as in the second sub-level of the level of spans. These features demonstrate the nesting of the previous level into the element level.

In real-world markup, the absence of tags for spans and elements, or the absence of both elements and spans in markup, is indicated using special marks. For instance, in the Human Values guideline, a special tag is used, "There are no values", to indicate that an element is labelled with no tags. Additionally, if a markup contains no elements or spans, assessors create only one element within the markup and label it with the mentioned tag.

Examples are given in Appendix B.

## 4.3 EXTENDED LEVEL

We have also extended the text markup format to include new entities and relationships at the level of individual elements. Six new entities have been added, as follows:

- *Assessor* creates markups of Documents
- *Dataset* is a set of marked up Documents
- *SpanOvertext* is a text string with a context information that characterize the span
- *SpanOvertextTag* is a label used to tag SpanOvertexts
- *ElementOvertext* is a text string with context information that characterize the element
- *ElementOvertextTag* is a label used to tag ElementOvertexts

We added the following new relations.

- *Assessor — Markup* The relation has the names: Assessor creates Markup, Markup is created by Assessor. The relation has the following features: Assessor can create 0 or more Markups; Markup may have only one Assessor-creator
- *Dataset — Document* Relation has the following names: Dataset includes Document, and Document belongs to Dataset. The relation has the following features: The dataset can include zero or more Documents; The document must belong to one specific Dataset
- *ESpan — the SpanOvertext* Relation has the following names: ESpan includes SpanOvertext, and SpanOvertext belongs to ESpan. The relation has the following features: One SpanOvertext can belong only to one ESpan; Espan may include 0 or more SpanOvertexts
- *SpanOvertext — SpanOvertextTag* Relation has the names: SpanOvertext is labeled by SpanOvertextTag, SpanOvertextTag labels SpanOvertext. The relation has the following features: the same SpanOvertextTag can label 0 or more SpanOvertexts; SpanOvertext can be labeled by zero or more SpanOvertextTags (SpanOvertext can be without tags)
- *Element — ElementOvertext* Relation has the following names: Element includes ElementOvertext, and ElementOvertext belongs to Element. A Relation has the following features: One ElementOvertext can belong only to one Element; Element may include 0 or more ElementOvertexts
- *ElementOvertext — ElementOvertextTag* Relation has the following names: ElementOvertext is labeled by ElementOvertextTag, ElementOvertextTag labels ElementOvertext. Relation has the following features: the same ElementOvertextTag can label o or more ElementOvertexts; ElementOvertext can be labeled by 0 or more ElementOvertextTags (ElementOvertext can be without tags).

We further define Dataset as a distinct entity, which includes metadata such as the creators of the original markups. Additionally, we introduce the concept of an "overtext" to handle complex, context-dependent annotations. This is exemplified by the Human Values Rink et al. (2024); Maysuradze et al. (2024) and UP Great READ//ABLE Levikin et al. (2025) datasets.

Examples are given in Appendix C.

## 4.4 CORRESPONDENCE BETWEEN TASKS AND HIERARCHY LEVELS

As the continuation of the extensive literature review analysis, we came up with a correspondence between the task types and the levels of the proposed hierarchy of models. Table 1 shows the composition of the relationships between tasks, datasets, and hierarchy levels. Also, for each combination of dataset and task, we report the number of publications in which this combination is mentioned. The good coverage of the publications allows us to conclude that the proposed system of data models is widely applicable.

## 4.5 FUNCTIONALITY OF RELATIONS IN THE CONCEPTUAL MODEL

When mapping conceptual models to logical implementations, such as relational or dictionary-based models, functional relationships play an important role. These relationships are unambiguous and

Table 1: The correspondence of datasets, task types, and the proposed hierarchy levels. Column 'Paper Count' reports the number of publications in which this combination is mentioned.

| Dataset | Task Type | Level | Paper Count |
|---|---|---|---|
| Kaggle NER Corpus (Bos et al., 2017) | NER | Span Level-1 | 25 |
| | POS Tagging | Span Level-1 | 5 |
| MultiCoNER (Malmasi et al., 2022) | NER | Span Level-1 | 85 |
| RuTermEval Dialogue (Mamontova et al., 2025) | NER | Span Level-1 | 8 |
| SWDA (Stolcke et al., 2000) | Dialogue Act | Span Level-2 | 228 |
| RuSentNE (Golubev et al., 2023) | NER | Span Level-1 | 5 |
| | Semantic analysis | Span Level-2 | 5 |
| | Opinion tuple extraction | Element Level | 2 |
| ADE (Gurulingappa et al., 2012) | Text classification | Span Level-1 | 10 |
| | NER | Span Level-1 | 150 |
| | RE | Element Level | 43 |
| | Coreference resolution | Element Level | 1 |
| DDI corpus (Herrero-Zazo et al., 2013) | NER | Span Level-1 | 102 |
| | RE | Element Level | 131 |
| PcMSP (Yang et al., 2022) | Text classification | Span Level-1 | 1 |
| | NER | Span Level-1 | 5 |
| | RE | Element Level | 5 |
| ChemProt (Krallinger M., 2017) | Text classification | Span Level-1 | 1 |
| | NER | Span Level-1 | 3 |
| | RE | Element Level | 70 |
| NERRE (Dagdelen et al., 2024) | NER | Span Level-1 | 10 |
| | RE | Element Level | 11 |
| NEREL (Loukachevitch et al., 2023) | NER | Span Level-1 | 9 |
| | RE | Element Level | 2 |
| | Extraction of multi-spans with named entities | Element Level | 1 |
| RURED (Gordeev et al., 2020) | NER | Span Level-1 | 3 |
| | RE | Element Level | 6 |
| | Entity linking | Element Level | 1 |
| Scierc (Luan et al., 2018) | NER | Span Level-1 | 137 |
| | RE | Element Level | 134 |
| | Coreference resolution | Element Level | 113 |
| CONLL 2012 Ontonotes (Pradhan et al., 2013) | NER | Span Level-1 | 87 |
| | Semantic Role labeling | Element Level | 25 |
| | Coreference resolution | Element Level | 15 |
| RuSuperGLUE's RWSD task (Shavrina et al., 2020) | Relation classification | Element Level | 7 |
| MERA's RWSD task (Fenogenova et al., 2024) | Relation classification | Element Level | 5 |
| MERA's Ruethics task (Pradhan et al., 2013) | Relation classification | Element Level | 1 |
| SemEval 2010 Task 8 (Hendrickx et al., 2019) | Relation classification | Element Level | 8 |
| SemEval-2018 Task 7 (Buscaldi et al., 2017) | NER | Span Level-1 | 14 |
| | RE | Element Level | 57 |
| | Relation classification | Element Level | 57 |
| | Knowledge Graph Construction | Element Level | 3 |
| UpGreat READ//ABLE (Levikin et al., 2025) | Extraction and classification of spans with errors | Span Level-2 | 2 |
| | Extraction of multi-spans with errors | Element Level | 1 |
| | Annotation of text spans with comments | Extended Level | 1 |
| Human Values dataset (Rink et al., 2024) | Text classification | Span Level-2 | 1 |
| | Extraction of spans with human values | Span Level-2 | 2 |
| | Extraction of elements with human values | Element Level | 2 |
| | Semantic analysis of elements with human values | Element Level | 2 |
| | Annotation of text markups with comments | Extended Level | 2 |

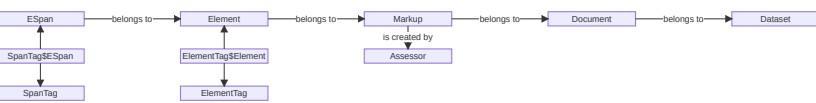

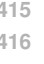

Figure 2: Functional relations in the proposed models.

universally defined, meaning that each instance of the first entity corresponds to exactly one instance of the second entity. For simplicity, let us consider an expanded conceptual model with no overtexts or tags in this section.

The extended conceptual model adds two entities ElementTag$Element (entity relation between ElementTag and Element) and SpanTag$ESpan (entity relation between SpanTag and ESpan) and can be described using only functional relations (see Fig. 2).

The proposed model includes two key components that play a significant role in content analysis and visualization: the snowflake schema. They can be distinguished by removing ESpan, SpanTag,

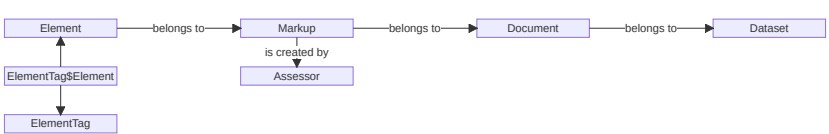

Figure 3: Snowflake schema with the tree root at ElementTag$Element.

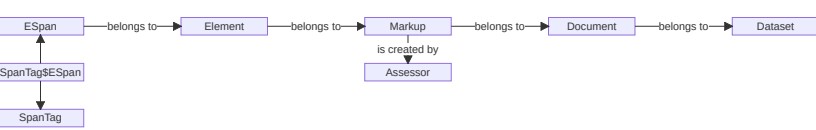

Figure 4: Snowflake schema with the tree root at SpanTag$ESpan.

SpanTag$ESpan or ElementTag$Element, ElementTag from the model as shown in the figure. 3 and fig. 4.

If we talk about the full conceptual model described above, then we need to only add overtexts that have a functional dependency on ESpans and Elements and then connect them with their corresponding tags using entity relations.

These properties of the extended conceptual model allow for a direct mapping to a logical relational model and a JSON model in a relatively straightforward manner. However, the solution is not unique.

### 4.6 LOGICAL RELATIONAL MODEL

There are various techniques to create logical models from conceptual models. This subsection will discuss a simple process of mapping the extended markup conceptual model to a logical relational database model. The mapping relies on standard implementations of functional dependencies using foreign keys and introduces attributes for entities.

This logical model is based directly on the conceptual model that includes functional relationships. It adds attributes to entities and their relations. The diagram of the logical model is provided in Fig. 5.

An important feature of the logical model is that all the foreign key fields are required (not null), which is widely regarded as best practice when designing logical models.

12 tables correspond to 12 entities mentioned above for the conceptual model. Their more detailed descriptions are given in Appendix E.

There are also 4 decoupling tables that correspond to entity relations with the same names. For example, in table SpanTag$ESpan there are two fields span_tag_id and span_id. The primary key consists of these two fields, which refer to the SpanTag and ESpan instances having a connection. Tables ElementTag$Element, SpanOvertextTag$SpanOvertext, and ElementOvertextTag$ElementOvertext are similar.

### 4.7 CLARIFICATIONS REGARDING THE TAGS

Certain aspects and constraints cannot be sufficiently captured within the diagrams above. First, within a single dataset, all SpanTag entities must possess unique names. This uniqueness constraint is equally applicable to other tag-like entities within the model. Second, while tags from different datasets may share identical names, this lexical coincidence does not imply semantic equivalence. This principle likewise extends to other tag-entities.

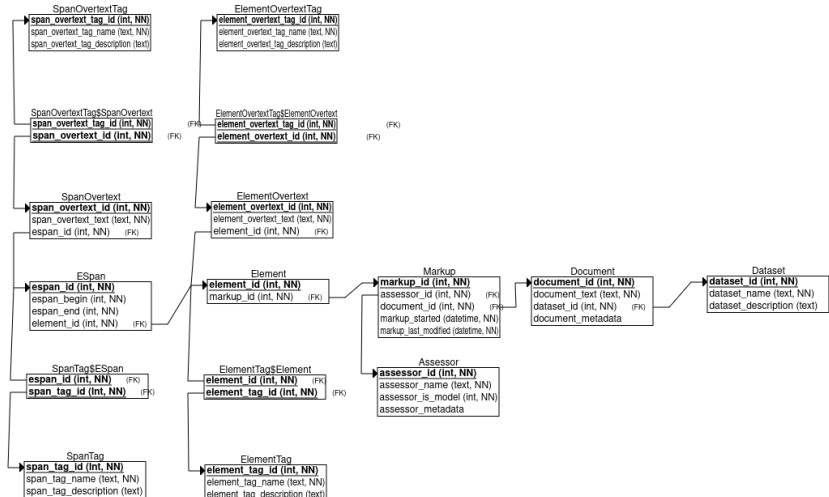

Figure 5: Logical relational model. Legend: *bold font* — primary key, *NN* — not null, *int* — integer data type, *text* — string data type, *FK* — foreign key.

## 5 CONCLUSION

This study introduces a system of nested data models, delineating their constituent entities and relationships. We define a novel hierarchical framework comprising several levels: the first and second sub-levels of the spans level, the element level, and an extended level. Within this framework, we demonstrate the functionality of the underlying relational structures and propose a new relational schema, complete with entity attributes. The inherent features and limitations of the models are also delineated.

To validate the broad applicability of this nested model system, we have developed a scalable, automated pipeline for identifying common research tasks associated with text annotation. A principal advantage of this pipeline is its extensive coverage, scalability, and multi-task capability, achieved without dependence on large panels of domain experts. An analysis of over a thousand publications enabled the identification of 17 distinct task types. A collection of tasks from 21 diverse datasets was subsequently mapped onto the proposed hierarchy, and a comparative analysis of their features was conducted to inform the selection of optimal AI applications.

The primary significance of the proposed framework lies in its demonstrable adaptability. It is capable of accommodating datasets with distinctive and complex features — such as the hierarchical tags present in the Human Values and UP Great datasets. A secondary justification is the convenience it offers to annotators during markup operations. The successful combination of flexibility and ease of use underscores the framework's considerable potential for broader application and future development.

## REPRODUCIBILITY STATEMENT

The robustness of the argument for the model's wide applicability rests on a substantial evidential base: a systematic review of a wide range of datasets, each connected to a variety of tasks referenced in multiple papers. Therefore, as for the reproducibility, the main points relate to the semi-automated literature analysis pipeline. All components of this pipeline, including the prompt used for a large language model, are described in the Appendix F.

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

## A   LEVEL OF FRAGMENTS EXAMPLES

### A.1   NER

Here, we demonstrate that the Kaggle NER Bos et al. (2017) corpus is represented at the span level (the first sub-level). In this example and other examples, we will be considering datasets with varying levels of complexity.

For the document

"Thousands of demonstrators have marched through London to protest the war in Iraq and demand the withdrawal of British troops from that country."

The markup of this document contains the following spans: "London", "Iraq" (labeled with the SpanTag "geo"), and "British" (labeled with SpanTag "gpe").

### A.2   MULTI-LABEL CLASSIFICATION AND SENTIMENT ANALYSIS

Here we demonstrate that RuSentNE Golubev et al. (2023) corresponds to the second sub-level of the span level. A document from RuSentNE Golubev et al. (2023) contains the following text:

"The case of El Salvador seems to have been a wake-up call for the United States, provoking a harsh response aimed at deterring other countries from taking similar actions," says Bonnie Glazer, an expert at the Center for International and Strategic Studies.

Its markup includes two spans. The first span is "expert at the Center for International and Strategic Studies". This ESpan is labelled with "PROFESSION" and "NEUTRAL" SpanTags. The second span is "El Salvador" and is labelled with "COUNTRY" and "POSITIVE". In this example, each span is labelled with a specific entity SpanTag and sentiment SpanTag.

## B   LEVEL OF ELEMENTS EXAMPLES

### B.1   RELATION EXTRACTION

An illustration from a NEREL Loukachevitch et al. (2023) document is below.

"The state of Colorado has completely abolished slavery In Colorado, 65% of voters voted for a total ban on slavery and forced labor. Previously, it was allowed as a form of criminal punishment. Back in 1865, the Thirteenth Amendment to the U.S. Constitution was passed, which prohibited slavery and forced labor, except as punishment for a crime. In the same year, it was ratified by 27 states, which meant that it entered into force and was valid throughout the United States."

Its markup includes an element that is labelled with the element tag "LOCATED_IN". This element includes two spans: "Colorado" (labeled with the span tag "STATE_OR_PROVINCE") and "United States" (labelled with the span tag "COUNTRY"). In this example, an element is the relationship between two ESpans.

### B.2   MULTI-FRAGMENT

We introduce examples of multi-fragments Maysuradze et al. (2024) from the Human values dataset Rink et al. (2024) and UpGreat READ//ABLE Levikin et al. (2025). Amongst the mentioned datasets only these data collections are not publicly available; nevertheless, we are eager to reference them as they remarkably support our further speculations.

An Element from Human Values dataset is labelled by tags "Positive tonality" and "Culture (norms) of behavior", and it has the 4 following ESpans:

- ESpan "In addition to preserving, studying and exhibiting cultural values, an important task of museums is to replenish the museum fund."

- ESpan "Systematic joint activities of departments to replenish museum funds are carried out traditionally, continuously and effectively"

- ESpan "donated approximately 1,000 items of cultural value."

- ESpan "the collections of nine federal museums have been replenished"

These multifragment spans work together to help us understand the author's thoughts.

An Element from UpGreat READ//ABLE Levikin et al. (2025) has the 4 following ESpans

- ESpan "During the Foreign campaigns (1813-1815), when Europe was being liberated from Napoleon, the Russian army saw that the standard of living of Europeans was higher, and they attributed this to the absence of absolute monarchy and serfdom in a number of European countries" is labelled by SpanTag "Cause"

- ESpan "Nicholas 1, who began to pursue a policy aimed at tightening the state regime." is labelled by SpanTag "Consequence"

- ESpan "Alexander 1 pursued a reactionary policy and refused to carry out liberal reforms in the second half of his reign." is labelled by SpanTag "Cause"

- ESpan "Revolutionary ideas began to spread, and various circles were created to discuss the shortcomings of the country, such as the Petrashev circle." is labelled by SpanTag "Consequence"

Thus, the multifragment spans take on different roles in various scenarios.

## C  EXTENDED LEVEL EXAMPLES

### C.1  MULTI-ASSESSOR TEXT MARKUPS

Document text from Human Values Dataset (Rink et al., 2024) that was translated from Russian to English:

*The Russian Ministry of Culture has transferred 90 confiscated cultural valuables to the Tsarskoye Selo State Museum-Reserve. The Northwestern Federal District Office of the Russian Ministry of Culture, by order of the Russian Ministry of Culture, has transferred confiscated and court-ordered state-owned items from the 19th and 20th centuries, recognized as cultural valuables, to the Tsarskoye Selo State Museum-Reserve for permanent safekeeping. "In addition to preserving, studying, and exhibiting cultural treasures, an important task for museums is to expand their collections. One way to do this is by legally transferring cultural treasures to state ownership, identified by law enforcement agencies and customs services during smuggling attempts and other offenses," said Alla Manilova, State Secretary and Deputy Minister of Culture of the Russian Federation. "Systematic joint efforts between agencies to replenish museum collections are traditionally, continuously, and effectively conducted." Thanks to her, the Russian Ministry of Culture transferred approximately 1,000 items of cultural value to federal museums in 2021. For example, in the Northwestern Federal District alone, the collections of nine federal museums were replenished by order of the Russian Ministry of Culture.*

This document has two different markups.

- The first markup was created by the first assessor and includes an element labelled with the ElementTags "legal consciousness" and "positive tonality", with the ESpan "legally transferring cultural treasures to state ownership".

- The second Markup is created by the second Assessor and includes an element labelled with the ElementTags "culture and art" and "neutral tonality", but with the ESpan "In addition to preserving, studying, and exhibiting cultural treasures, an important task for museums is to expand their collections. One way to do this is by legally transferring cultural treasures to state ownership".

This example shows that two assessors labelled elements that contain different ESpan's.

## C.2 OVERTEXT

The ESpan comes from the UpGreat dataset (Levikin et al., 2025). It has been labelled with the tags "error" and "CONFIRMATION_ERROR". The text it contains is as follows:

"An example from my own experience is the Unified State Exam. Eleventh graders are already imagining what it will be like to take the exam and what types of questions they will face in the KIMS. They are preparing for it, and this is an example of rational thinking. However, without experiencing the exam first-hand, students can only imagine what they will go through."

This ESpan includes an explanation of the error, which is referenced in the SpanOvertext.

*SpanOvertext*: The example is presented in everyday language and does not distinguish between two types of cognitive processes. The first type, observed-perceptual approximation, is not clear about the methods used. The second type, understood on a rational level, does not explain how this understanding was formed based on the object of knowledge.

Thus, this context information in the Overtext structures from UpGreat READ//ABLE (Levikin et al., 2025) helps us understand tutors' reasoning.

# D TRANSITIONS BETWEEN HIERARCHY LEVELS

For each task mentioned above, we can determine the minimally sufficient hierarchy level in the proposed system of data models by which it can be represented adequately. In data modelling literature this minimally sufficient level may be referred to as 'parsimonious'.

If necessary, a transition to a higher level may be offered as follows.

It doesn't prevent the presentation of these datasets on higher levels because of nesting. In this section, we will discuss standard transitions from lower to higher levels.

## D.1 FROM THE FIRST SUB-LEVEL OF SPAN LEVEL TO THE SECOND SUB-LEVEL

In this transition, the instances of entities ESpan, SpanTag, and Markup, as well as the relations between them, do not change. Let $D$ be a set of instances of Document, $M$ be a set of markups, $text(d)$ be a text string of document $d \in D$, $markup(d) \in M$ be a single markup of document $d \in D$. On set $D$ we introduce the equivalence relation $d_1 \sim d_2 \Leftrightarrow text(d_1) = text(d_2)$ where the last equality is a character-by-character equality between strings. Based on this relation, we introduce equivalence classes on D: $Classes = \{C_{text} : C_{text} = \{d \in D : text(d) = text\}\}$.

When moving to the second sub-level, an instance $d_{text}$ of the Document entity is created for each equivalence class $C_{text}$, with the corresponding text. This document is then associated with the set of markups of documents from the equivalence class, i.e. $markups(d_{text}) = \{markup(d) : d \in C_{text}\}$. And on the second sub-level of the span level we consider documents $D_{new} = \{d_{text} : C_{text} \in Classes\}$ where document $d_{text} \in D_{new}$ has markups $markups(d_{text})$.

Thus, when moving to the second sub-level, we propose to consider documents with the same text as a single document to support multi-assessorship. At the same time, the structure of the markups themselves is preserved.

## D.2 FROM SPAN LEVEL TO LEVEL OF ELEMENTS

Let $S$ be a set of instances of entity ESpan on span level, $markup(s)$ is the markup that includes this span $s \in S$, $tags(s)$ is the set of span tags labelling $s \in S$. In this transition we assign each span $s$ to an element $e = element(s)$. This element in the new data format will have the following relations: $e$ includes $s$, $e$ belongs to $markup(s)$. In addition, instances of the "Document" entity do not change during the transition, but their markups now include created instances of the element entity.

Thus, we wrap fragments in elements. And now we consider not spans but elements identified with them.

## D.3 From element level to extended level

In this process, a markup is assigned to an "UNKNOWN ASSESSOR," and each document is assigned to a Dataset instance.

## E Attributes of entities

We provide lists of attributes for each of the 12 tables corresponding to the entities from the conceptual models.

- *Dataset* Contains the primary key dataset_id and fields dataset_name and dataset_description
- *Document* Contains the primary key document_id, field document_text and a foreign key dataset_id referring to the corresponding Dataset
- *Assessor* Contains the primary key assessor_id and fields assessor_name and assessor_is_model (a flag indicating that the Assessor is a model or a person)
- *Markup* Contains the primary key markup_id, a foreign key document_id (referring to the corresponding document), and a foreign key assessor_id (referring to Assessor). If the original markup contains no information about the assessor, then, to avoid the presence of null foreign keys, the field assessor_id refers to the id of an assessor called "Unassigned Assessor".
- *Element* Contains the primary key element_id and a foreign key markup_id referring to the corresponding Markup
- *ESpan* Contains the primary key espan_id, fields espan_begin and espan_end (borders of span in Document's text) and a foreign key element_id referring to the corresponding Element
- *ElementTag* Contains the primary key element_tag_id and fields element_tag_name and element_tag_description
- *SpanTag* Contains the primary key span_tag_id and fields span_tag_name and span_tag_description
- *SpanOvertext* Contains the primary key span_overtext_id, field span_overtext_text and foreign key espan_id referring to the corresponding ESpan
- *SpanOvertextTag* Contains the primary key span_overtext_tag_id and fields span_overtext_tag_name and span_overtext_tag_description
- *ElementOvertext* Contains the primary key element_overtext_id, a field element_overtext_text and a foreign key element_id referring to the corresponding Element
- *ElementOvertextTag* Contains the primary key element_overtext_tag_id and the fields element_overtext_tag_name and element_overtext_tag_description

In addition, the Document and Assessor tables may have metadata fields that can vary.

## F Extensive Review Pipeline

### F.1 Goals and objectives

To demonstrate the broad applicability of our nested data models, we proposed the following pipeline for compiling a list of common research tasks involving text annotation. Note that the article clearly states that different tasks could be solved on the same dataset in different studies.

We decided that in order to clearly divide tasks, the task description should contain the following characteristics:

- Input data
- Output data

- Correspondence to the level in the nested data models (this point can be directly deduced from the previous two)

In addition, we identified the following characteristics for creating a benchmark of machine learning models for solving generalized text annotation tasks in future work, but we do not discuss these issues in this article:

- Quality criteria
- Machine learning models used to solve the task, as well as their comparison using the quality criteria specified in point 4.

## F.2 PIPELINE CHOICE

The pipeline had to meet the following requirements: broad coverage to demonstrate that our nested model system can address the widest possible range of tasks; scalability, since the breadth of coverage may require reviewing thousands of articles; the ability to delegate to less qualified personnel (e.g., undergraduate students in relevant fields), since despite the scalability requirement, such a broad review still requires a large amount of resources.

The search for a pipeline that meets all of the above characteristics has not been successful, so we propose our own literature review pipeline that meets the specified requirements.

## F.3 PIPELINE DESCRIPTION

First, we expertly selected tasks that are common in scientific research and most clearly relate to our system of nested data models: NER and RE. We analyzed existing reviews of datasets for these tasks. These studies helped us select datasets for this research. Since our future goal is to create a generalized text annotation benchmark, we tried to select datasets from different scientific fields to maximize the breadth of the study, while choosing a minimum representative number of datasets so that evaluating the machine learning model on the benchmark would not take up too many resources.

After forming a list of datasets, we reviewed the literature for each of them as follows:

1. Searching for the original article about the dataset and extracting the following information from it:
   - General description of the dataset
   - Description of the task for which the authors created the dataset (the format of the task description was given above)
   - Dataset license
2. Generating a list of articles that reference the original using the SemanticScholar API, downloading them, and converting them to plain text. This step was fully automated using a Python script.
3. For each of the downloaded articles, retrieving information about the tasks that the authors solved on the dataset in their study from the article text. This step was fully automated using an LLM-based workflow: the text of each article was given to the LLM with a special prompt (the prompt can be found in the next subsection of this appendix), and the LLM output was logged to a separate file.
4. Calculating statistics based on the LLM-analysis: the total number of articles analyzed and the distribution of articles by the tasks being solved. If plain text could not be obtained for an article, it was not included in any of the results of our study. Articles that refer to the original article on the dataset but do not explicitly describe a task solved on the dataset (LLM returned no results) are classified as articles in which only the task proposed by the authors of the dataset was being solved.

The experiments reported in this paper are conducted using the Qwen3-235B-A22B-Instruct-2507 model, selected due to its public availability and sufficiently strong performance. We emphasize, however, that the proposed framework does not rely on model-specific characteristics; any sufficiently large and instruction-tuned language model can be integrated into the pipeline without loss of generality.

## F.4   LLM PROMPT

The LLM prompt used for retrieving information about tasks from the article text is given below. `DATASET_NAME` and `ARTICLE_PLAIN_TEXT` are provided for each dataset.

**LLM Prompt Text**

*Summarize the following scientific article by extracting information about the problems solved on the DATASET_NAME dataset. For each problem, extract the following details:*

1. *Input data: Describe what is fed into the model's input.*
2. *Output data: Describe what is expected as the model's output.*
3. *Quality criteria:*
   - *Technical: Specify the criteria used by researchers or mathematicians to evaluate how well the problem has been solved.*
   - *Humanitarian/business: Specify the criteria used by domain experts (e.g., healthcare professionals or pharmaceutical companies) to evaluate how well the problem has been solved.*
   - *Tolerable error: Define the level of error that is acceptable for the problem to be considered well-solved.*
4. *Models used: List the models employed to solve the problem and compare them according to the above-mentioned quality criteria.*

*If any of these parts are not mentioned, only return the number of the missing part(s) without additional information.*

*Do not write names of the paragraphs, only write their numbers.*

*If the article does not mention any problem related to the DATASET_NAME, return the message: "NO INFORMATION".*

*Do not include background information, methodology details, or future work. Only state the requested facts.*

*Article: ARTICLE_PLAIN_TEXT*

