# OpenReview forum: "Nested Text Labelling Structures to Organize Knowledge in AI Applications for the Humanities and Social Sciences"
_ICLR.cc/2026/Conference — ICLR 2026 Conference Withdrawn Submission_

### Official Review · Reviewer_5jBS · 2025-10-30

**Soundness:** 1
**Presentation:** 2
**Contribution:** 1
**Rating:** 2
**Confidence:** 4

**Summary:**

- The paper introduces a unified data model/schema for text annotations
- The schema consists of three levels
    - Span Level: Highlight text fragments and tag them (eg. NER, POS, etc)
    - Element Level: Group multiple related spans into elements that represent relationships or structures (more complex tasks like Relation Extraction (RE), Coreference Resolution, and Frame Extraction)
    - Extended level: Add metadata like who annotated it, comments explaining decisions, and context notes on spans/elements (eg. context-dependent annotation such as human values dataset)

- Validation of the schema
    - The authors validate their framework's usefulness and broad applicability by demonstrating it can successfully model and categorize a wide range of existing, real-world annotation tasks and datasets.
    - 21 diverse datasets (including CONLL 2012, ADE, and RuSentNE) were analyzed by reviewing "over a thousand" associated publications found via the Semantic Scholar API.
    - Used LLM to extract what tasks each paper addresses (Identified 17 distinct task types)
    - Map all these combinations of datasets and tasks to their proposed 3-level hierarchy (Table 1 of the paper)

**Strengths:**

- The paper's main contribution is a new hierarchical framework to standardize text annotation (more details in the summary above)
- The paper claims to solve the "fundamental trade-off" that is an existing  problem text annotation acorss disciplines. The nested levels allow an expert to choose the simplest model possible for their task, while still providing the power to handle complex nuances (like multi-span elements or contextual comments) when needed.
- The paper provides complete formal specifications: ER diagrams, relational schemas, entity definitions which also handles edge cases such as multi-annotator, comments, context, etc.

**Weaknesses:**

- The paper claims that the current models fail to capture "nuances" but doesn’t really discuss what specific nuances are lost? They just assert this without proof. And the entire premise of the paper is based on existence of the problem, without a sufficient evidence that the problem exits
- There are claims that the schema is an attempt to solve "convenience vs expressiveness" trade-off but the paper never measures or provide evdience for how would it lead to better convenience (eg. user studies) or expressiveness (eg. ML experiments). In short there is no validation that the current schema would be useful.
- The mapping of 21 datasets to the schema, shows that datasets can be mapped to the schema. The pipelines used LLM-as-judge to annotate the type of tasks in the paper, however, there is no validation that the LLM-as-judge actually works or any robustness checks, making the pipeline and subsequent results unreliable.

**Questions:**

- I am unsure about the who exactly does the schema help. Is it for the annotators to better understand the LLM generated annotations or it for better annotations using LLMs?
- Why is a unified schema better (or more precisely why is having 3 levels better than having 1 flexible schema)? Different tasks might genuinely need different representations. Forcing everything into one framework could make things worse.

---

> ### Author Response · Authors · 2025-11-20
> **Comments to the Official Review of Submission21477 by Reviewer 5jBS**
>
> We thank the reviewer 5jBS  for their time and feedback.
> In the comments, we would like to clarify the queries below based on our understanding of the work.
>
> Q1. I am unsure about the who exactly does the schema help. Is it for the annotators to better understand the LLM generated annotations or it for better annotations using LLMs?
>
> Comments 1.
> On the one hand, domain experts must be able to formally represent their complex, structured knowledge; otherwise, the comprehensive training and validation of AI systems becomes unfeasible. (References to complex structures to be added.)
> On the other hand, contemporary experience in developing AI systems demonstrates that they must be oriented towards multiple heterogeneous tasks from the outset of their training. Foundation models serve as a prominent example of this paradigm.
>
> Q2. Why is a unified schema better (or more precisely why is having 3 levels better than having 1 flexible schema)? Different tasks might genuinely need different representations. Forcing everything into one framework could make things worse.
>
> Comment 2.
> The upper level of our architecture is, in itself, a flexible schema. Empirical evidence from AI-driven projects in the humanities indicates two key findings:
> 1.	Users (human experts) prefer task descriptions that operate at a minimal level of complexity.
> 2.	Contemporary technology functions more effectively with less complex descriptions.
>
> The assertion that "different tasks might genuinely need different representations" is undoubtedly valid when considering human interaction. It is therefore essential to employ models of reduced complexity within user interfaces and documentation.
> However, experience in AI system development has demonstrated the superiority of a multi-task approach during the training phase. Conversely, for deployment and to mitigate computational overhead, it is advantageous for the software solution to operate with a model of the minimally sufficient complexity.
> A thorough analysis of this trade-off is subject to ongoing investigation within our research; however, a comprehensive treatment of the matter falls beyond the scope of the present work.
>
> Thank you for mentioning the weaknesses to be accounted for too. We look forward to answering any further queries you may have and further discussions.

---

### Official Review · Reviewer_cmqx · 2025-11-02

**Soundness:** 1
**Presentation:** 2
**Contribution:** 1
**Rating:** 0
**Confidence:** 5

**Summary:**

The paper proposes a framework for performing knowledge management extracted from the textual information via annotation models. These can be extracted with the help of the tasks of entity and relation extraction.

**Strengths:**

Since the motivation and research questions are not described, it is hard to see the strengths of the paper.

**Weaknesses:**

--> There is no mention of humanities and social sciences in the rest of the paper other than the title.
--> The introduction is missing references to support the claims.
--> The exact research questions that the authors want to target are missing.
--> There is no related work.
--> After the generalized logical or relational model that is created with the NER and Relation extraction tasks, how is that useful in any of the downstream tasks?
--> The paper is missing the evaluation of the proposed knowledge management framework.
--> What is the broad impact of this work?

The paper is still in a premature stage and the contributions are not very clear.

**Questions:**

--> What are the research questions that authors are targeting?
--> After the generalized logical or relational model that is created with the NER and Relation extraction tasks, how is that useful in any of the downstream tasks?
--> How this framework can be evaluated?
--> What is the broad impact of this work?

---

> ### Author Response · Authors · 2025-11-20
> **Comments to the Official Review of Submission21477 by Reviewer cmqx**
>
> We thank the reviewer cmqx  for their time and feedback.
> In the comments, we would like to clarify the queries below based on our understanding of the work.
>
> Q1. What are the research questions that authors are targeting? + Q2. After the generalized logical or relational model that is created with the NER and Relation extraction tasks, how is that useful in any of the downstream tasks?
>
> Comments to Q1+Q2.
> It is widely acknowledged, particularly within the humanities, that text is perceived as "a complicated web of interwoven and overlapping relationships of elements and structures" (Vanhoutte, 2006). Consequently, modelling textual information results in multi-layered and non-linear objects (Bleeker, 2018). This understanding has, for many decades, driven the development of data models and file formats for representing textual markup. A partial history of this development, beginning with the Standard Generalized Markup Language (SGML, 1986), is detailed in Portier (2012).
> Recent advancements in Artificial Intelligence have demonstrated that substantial success in AI applications is achieved through the development of multi-task models trained on maximally diverse datasets. Consequently, AI researchers developing solutions for the humanities now face the challenge of creating models and datasets that simultaneously encompass different subject domains and a wide variety of tasks. A desirable goal is the conversion of numerous existing datasets into a common, universal format. However, this endeavour leads us directly to the next evolutionary stage of data modelling.
> Furthermore, over the years, humanities scholars have come to expect computational methods to answer research questions of increasing structural complexity. In response to this, datasets requiring the most structurally complex markup are still relatively scarce. Recent years, however, have witnessed the emergence of tasks in the humanities with complexly structured inputs and outputs, including state-level initiatives, for instance, in the field of school education automation.
> The key objective of our research is to develop a data model to which significantly diverse existing datasets from the humanities can be mapped. Subsequently, within this universal model, AI-based solutions can be validated and developed.
>
> Q3. How this framework can be evaluated?
>
> Comment 3.
> To empirically validate the proposed solution, we conducted a systematic large-scale review of scientific publications concerning diverse tasks and datasets. This review serves to validate the framework's universality by demonstrating that all tasks identified in the literature can be represented within it in a coherent and practicable manner.
> The creators of the aforementioned datasets and tasks with complex structured inputs and outputs had previously analysed existing formats and models, and found it necessary to devise their own novel solutions. This challenge, the absence of adequate models, is one we have also encountered in our own research. Consequently, it can be asserted that our proposed solution addresses a timely and pertinent research gap, thereby establishing its novelty.
> The optimal complexity of the proposed model is justified by its inherent multi-level architecture, which provides the necessary flexibility without introducing undue complexity.
>
> Q4. What is the broad impact of this work?
>
> Comment 4.
> To demonstrate the broad applicability of our framework beyond a niche application, we conducted an extensive literature review, analysing over 1,500 publications and identifying 17 distinct task types. This scope indicates that the proposed framework has the potential to be utilised by thousands of researchers working across a diverse range of problems.
> As previously noted, it is anticipated that the next generation of datasets will facilitate the development of novel AI models capable of addressing far more complex challenges within the humanities. The emergence of such models is poised to have a profound impact on the trajectory of humanities research.
>
> Minor weaknesses mentioned will be accounted as well. Thank you. We look forward to answering any further queries you may have and further discussions.

---

### Official Review · Reviewer_owiv · 2025-11-07

**Soundness:** 2
**Presentation:** 2
**Contribution:** 2
**Rating:** 4
**Confidence:** 2

**Summary:**

This paper proposes a hierarchical data model with three levels of increasing complexity. The motivation is that existing text annotation models often fail to capture fine-grained expert knowledge, and there is a need to balance convenience for annotators with the expressive power of the annotation representation. The authors develop a semi-automated literature review pipeline by querying the Semantic Scholar API and extracting relevant task descriptions using a large language model. Through this process, they analyze 21 datasets and identify 17 distinct types of text-labeling tasks. Based on these findings, they design a nested text-labeling framework that organizes these tasks according to their structural complexity. The authors highlight the adaptability of the framework to datasets with varying levels of complexity, as well as its practical convenience for human annotators. Finally, they demonstrate how the framework can be implemented as relational database structures and propose a corresponding relational schema.

**Strengths:**

1. Clear motivation for using a hierarchical model that introduces additional complexity only when necessary
2. Good coverage of identified annotation task types across multiple datasets.
3. Potential for unifying datasets and standardizing annotation practices
4. Helpful explanation of functional dependencies and how the proposed framework maps to relational database schemas

**Weaknesses:**

1. **Missing references:** I am afraid but there are very few references provided. For example:
   - Lines 43–44: "In the humanities and social sciences, recent studies suggest that current text annotation models often fail to encapsulate the full nuance of expert knowledge, thereby limiting their utility for advanced AI applications."
     This is your main motivation for proposing new data models, but no sources are cited to support this claim.
   - Line 223: "The data model described in this subsection is a traditional one and can be found in many different datasets."
     Please cite which datasets and sources.
   - Lines 80–81: "Existing methods proved inadequate, primarily due to limitations in scalability and coverage."
     Which methods?

2. **Missing related work section:** You do not have a related work section. You do position the contributions in the context of prior literature. What is the closest related work? How do other approaches model annotation structures? You rarely mention related work and do not contrast your work with existing models.

3. **Clearly state contributions:** I suggest including a dedicated paragraph that explicitly lists the paper’s contributions. For example, is the semi-automated literature review pipeline intended to be a main contribution as well?

4. **Significance of contributions:** Beyond the general motivation for improved annotation models, several contribution claims need to be moderated or substantiated:
   1. Lines 65–68 "The employment of the proposed model hierarchy during development, coupled with training on diverse task types, is posited to yield a new generation of machine learning models. These models would be characterised by greater usage simplicity and a broader range of application." This is a very strong statement. For a your proposed work of better organising annotation structures this claim seems a bit far-fetched.
   2. You emphasize convenience for annotators, but no empirical evaluation of annotation usability or efficiency is provided. When motivating your work with that and highlighting this in the conclusion section, please ensure you can substantiate that.
   3. You state that the structure “facilitates the selection of an appropriate AI model for any specific dataset” (lines 142–143). This may be true in principle, but for ICLR, this contribution is relatively limited unless comparative evidence is provided showing measurable improvements over existing data models.

5. **Validation:** The framework is derived from 17 identified tasks. You then conclude (lines 318–319) that the framework is widely applicable. To support this claim, I would argue to validate the system on tasks not used in the design process.

6. **Limitations:** You do not clearly articulate the paper's limitations. Which types of tasks or annotation scenarios are not well supported by the framework, and why? I would expect a dedicated section or paragrpah listing limitations.

7. **Semi-automated pipeline:** The pipeline uses an LLM, but the specific model is not identified. How do you ensure correct extraction? Was the output manually verified? This is particularly important since large models are known to struggle with “needle in the haystack” retrieval tasks (see: https://research.trychroma.com/context-rot).

8. **Vague**: Lines 18-19: "Experience gained through a number of applied projects and research studies has shown that there cannot be a single simple answer; [...]" That is very vague. Are these your own or do you draw this from the literature.

9. **Typos**:
	- "Analisys" in Figure 1
	- "can label o and more" in line 295

**Questions:**

- Extended Level is only covered by three publications. This seems quite low compared to the other levels?
- Line 470: "A collection of tasks [...]" this refers to the 17 distinct tasks?
- Where exactly do you refer to the AI applications in the Humanities and Social Sciences in the paper?

---

> ### Author Response · Authors · 2025-11-20
> **Comments on the Official Review of Submission21477 by Reviewer owiv**
>
> We thank the reviewer owiv for their time and feedback. In the comments, we would like to clarify the queries below based on our understanding of the work.
>
> Q1. Extended Level is only covered by three publications. This seems quite low compared to the other levels?
>
> Comment 1. The research design employed in this study involved an analysis of existing published datasets. We will incorporate references to the studies that advocate for the use of extended-level annotation frameworks, yet which lack accompanying published datasets.
>
> Based on these additional references, it can be argued that researchers have long recognised the necessity of enhancing the complexity of data structures in humanities research. However, it is only in recent years that large-scale datasets featuring such sophisticated structures have been systematically compiled and published. Consequently, the volume of publications utilising these complex datasets remains significantly lower than those relying on simpler data models.
>
> Q2. Line 470: "A collection of tasks [...]" this refers to the 17 distinct tasks?
>
> Comment 2. This refers to seventeen distinct types of tasks. For instance, the task of identifying a text fragment that an expert would classify as a Named Entity (NE) and assigning one specific tag to it is considered a single task type. It should be noted, however, that in the literature review conducted, tasks of this same type are encountered with different sets of labels and across different genres of text. The wording has been amended accordingly in line 470.
>
> Minor weaknesses mentioned will be accounted for as well. Thank you.
>
> We look forward to answering any further queries you may have and further discussions.

---

### Author Response · Authors · 2025-11-19
**Advanced labelling frameworks in the Humanities, given their considerable utility for tasks involving context, multi-assessor labelling, and the extraction of subjective textual categories**

We thank the reviewers owiv, cmqx, and 5jBS  for their time and feedback.
In the comment, we would like to clarify the common query raised by all the reviewers as below based on our understanding of the work.

Q1 by owiv + Weakness by cmqx: Where exactly do you refer to the AI applications in the Humanities and Social Sciences in the paper? There is no mention of humanities and social sciences in the rest of the paper other than the title.

Response: We would like to clarify the application and interpretation here.
A corpus of literature has emerged which successfully implements advanced annotation frameworks, incorporating multi-fragment and multi-assessor labelling protocols alongside contextual data. This approach facilitates the tagging of nuanced semantic errors in secondary-level school essays—for example, identifying fallacies in causal relationships—as well as the annotation of highly subjective constructs, including universal human values. The application of such rigorously defined, expert-driven annotation frameworks provides a foundation for developing machine learning models capable of performing automatic text markup. The quality of the annotation generated by these models has been shown to exceed the reliability of annotations performed by school teachers or individual subject-matter experts. It was this compelling potential that prompted our systematic review of existing annotation schemas to ascertain their standard applications across a range of specific tasks.
We concur that our emphasis on the application of complex annotation schemas to socio-humanities tasks was insufficiently articulated.
It is also pertinent to alert the ICLR community to the advent of these advanced frameworks, given their considerable utility for tasks involving context, multi-assessor labelling, and the extraction of subjective textual categories.
We hope the above comments help clarify the point. Thank you for your time and consideration.

We therefore will incorporate further arguments on this point and additional references, including within the main body of the Related Work section.

We are goind to reply to your other points in subsequent comments.
We look forward to answering the other questions as well as any further queries you may have and further discussions.

---

### Author Response · Authors · 2025-12-04
**Combined comment to the Chairs**

We thank the Chairs and reviewers owiv, cmqx, and 5jBS for their feedback. Our revisions, summarised below, address all concerns raised. Detailed point-by-point responses to reviwers’ questions were provided in the previous comments; here we aggregate key amendments.

Title & Scope: We concur the proposed knowledge representation applies beyond the humanities/social sciences. So we remove this disciplinary reference from the title. Simultaneously, in the Related Work section, we established that complex markup structures are shown to be particularly effective within humanities-oriented tasks, such as the automated assessment of secondary-level history essays or the identification of subjective textual categories, including universal human values.

Motivation & Literature: We have substantially revised the "Related Work" section to better motivate the research, citing literature that documents a decades-long trend toward increasingly complex text markup formats and data models.

Common Q1 (owiv + cmqx): Where exactly do you refer to AI applications in the Humanities/Social Sciences?We acknowledge our initial emphasis was insufficient. A key driver for this work is the emergence in Digital Humanities of advanced annotation frameworks (using multi-fragment/assessor protocols) for tasks like tagging semantic errors in essays or subjective constructs like values. These expert-driven frameworks provide the foundation for training high-quality automatic markup models. Our systematic review of schemas was prompted by this potential.

owiv Q1: The 'Extended' level is covered by only three publications.This reflects the current state of published datasets. While the need for complex data structures in humanities research is long recognised, large-scale datasets implementing them are a recent development. We will add references to studies advocating for such frameworks despite lacking public datasets.

owiv Q2: Clarification of "a collection of tasks".This refers to 17 distinct types of tasks (e.g., assigning a single Named Entity tag). Each type may appear in the literature with different label sets or across genres. The wording has been clarified.

cmqx Q1+Q2: What are the research questions and how is the generalized model useful?Text in the humanities is a "complicated web" of overlapping structures, necessitating multi-layered data models. Concurrently, AI progress relies on multi-task models trained on diverse data. Thus, a core challenge is creating models/datasets that span domains and tasks. Our key objective is to develop a universal data model to which diverse humanities datasets can be mapped, enabling the development and validation of AI solutions within a unified framework.

cmqx Q3: How is this framework evaluated?We validate universality via a systematic large-scale review, demonstrating all identified tasks can be coherently represented within our model. The necessity for novel solutions reported by creators of complex datasets confirms a pertinent research gap, which our model addresses. The multi-level architecture provides optimal, justified flexibility.

cmqx Q4: What is the broad impact?Our review of over 1,500 publications and 17 task types indicates potential utility for thousands of researchers across diverse problems. The framework facilitates the next generation of datasets and AI models for complex humanities challenges, poised to significantly impact the field.

5jBS Q1: Who does the schema help—annotators or LLM training?Both. Domain experts require formal representations of complex knowledge to train/validate AI systems. Simultaneously, AI systems must be oriented toward multiple heterogeneous tasks from the outset (as seen in foundation models). The schema serves this dual need.

5jBS Q2: Why is a unified three-level schema better than one flexible schema?Our upper level is a flexible schema. Evidence shows: 1) human experts prefer minimal-complexity descriptions, and 2) technology often works better with less complex descriptions. While different human-facing tasks need different representations, multi-task AI training benefits from a unified framework. The trade-off for deployment efficiency is an ongoing research question beyond this paper's scope.

We have incorporated these points and additional references into the submission. We contend that our contribution to knowledge representation holds an interest for the ICLR community, and we anticipate that the current version merits acceptance for both publication and presentation at this prestigious forum.

---

### Note · Authors · 2026-07-20

I have read and agree with the venue's withdrawal policy on behalf of myself and my co-authors.

---

### Meta-Review · Area_Chair_42L9 · 2026-01-06

**Summary:**

The three reviewers raised several critical concerns, including: the lack of a clear connection to AI applications in the humanities and social sciences beyond the title; insufficient motivation and missing citations to support key claims; the absence of a related work section; vaguely defined research questions and contributions; no empirical evaluation or validation of the proposed framework's utility, convenience, or impact.

**Reviewer Concerns:**

The authors' rebuttal partially addressed some of the concerns, such as clarifying the link to humanities tasks and adding a related work section. However, major outstanding concerns remain unresolved. For example, there is still no empirical evidence or user studies validating the framework's convenience, expressiveness, or superiority over existing models.

**Reviewer Scores:**

Reviewer owiv (initial score: 4 - weak reject): Given that the rebuttal addressed some clarifications (like task types and humanities context) but did not resolve the fundamental issues around lack of validation and unsubstantiated claims, it is unlikely the score would improve significantly.

Reviewer cmqx (initial score: 0 - strong reject): This reviewer's core concerns about missing research questions, evaluation, and impact were met with conceptual justifications rather than new evidence or experiments. So the score would almost certainly remain a strong reject.

Reviewer 5jBS (initial score: 2 - reject): The rebuttal engaged with their questions about the schema's users and the three-level design but did not provide the missing validation (user studies, ML experiments) or evidence that the "convenience vs. expressiveness" trade-off is actually improved. So the score would probably still be 2.

---

### Decision · Program_Chairs · 2026-01-26

Reject